# Environmental Factors Determining the Chorology of *Fraxinus angustifolia* in the Region of Murcia (Southeastern Spain)

**DOI:** 10.3390/life15020163

**Published:** 2025-01-24

**Authors:** Alfonso Albacete, Miguel Ángel Sánchez-Sánchez

**Affiliations:** 1Institute for Agroenvironmental Research and Development of Murcia, E-30150 Murcia, Spain; 2Center of Studies in Geography and Spatial Planning, 3004-530 Coimbra, Portugal; 3Department of Geography, University of Murcia, E-30001 Murcia, Spain; miguelangel.sanchez2@um.es

**Keywords:** ash tree, aridity, climate change, riverbed

## Abstract

The narrow-leaf ash (*Fraxinus angustifolia*) is distributed across southern Europe and northern Africa. Descriptions from 1751 report the presence of a large number of these trees in the Maritime Department of Cartagena; however, their numbers are actually greatly reduced in the Region of Murcia. This species is protected and finds refuge in riverbeds within the semi-arid environment of southeastern Spain. The desertification affecting this area of continental Europe has a significant impact on natural systems, with the ash being particularly vulnerable due to its water requirements. The objective of this study was to identify the locations of individual ash trees and assess the conditions of their surrounding environment. A literature review was conducted, and based on the environmental conditions necessary for their survival, the entire region was surveyed using GPS. A total of 670 trees were geolocated. The majority are situated in riverbeds. They are absent from the southern coastal zone and at higher altitudes above 1000 m above sea level in the Region of Murcia. Monitoring its reduction, expansion, and development could provide insights into how climate change is evolving and the potential extinction of this species in arid and semi-arid regions.

## 1. Introduction

The ash tree (*Fraxinus* spp.) is distributed across central–southern Europe and parts of northwestern Africa. The taxon *Fraxinus angustifolia* subsp. *angustifolia* is found in southwestern Europe and Africa, while *Fraxinus excelsior* is more characteristic of the northern regions [1,2] (Figure 1). Regarding the distribution of the ash tree in the Iberian Peninsula (Spain, Portugal, and Andorra), it is present throughout much of the peninsular territory. In the distribution of *Fraxinus angustifolia* in Europe and Africa (Figure 1), this species is considered “non-native” and absent in the southeastern Iberian Peninsula, the Levantine coast of Spain, and parts of southern Spain. This contrasts with historical forest conditions in the Region of Murcia, as documented in a 1751 report by the Maritime Department of Cartagena about the forest territory of the Region of Murcia [3]. In this report, “the forest inspector of the Navy, Juan Francisco de la Torre,” describes the “abundance of oaks, holm oaks, cork oaks, pines, poplars, hackberries, ash trees, and walnut trees” [4].

In the Region of Murcia, this species is absent in areas closer to the coast, a small portion of the southeastern Iberian Peninsula within the Spanish Mediterranean arc [2]. In Spain, *Fraxinus angustifolia* extends across all provinces with a Mediterranean climate, appearing sporadically in forests, groves, and riverbanks (Figure 2 and Figure 3) [5]. Other authors claim that the species is present in groves, riverbanks, and forests throughout all provinces, though it is scarce or absent in the arid areas of the Southeast, the Cantabrian Cornice, and the mid-to-high regions of the Pyrenees [6]. Additionally, other researchers, in a review of *Fraxinus angustifolia* groves in the south and west of the Iberian Peninsula, confirm the presence of this taxon in southeastern Spain [7].

Some maps depicting the distribution of *Fraxinus angustifolia* in Spain indicate the presence of this taxon within the Region of Murcia, specifically in two locations [8]. Other sources suggest a broader territorial distribution of the species in the Region of Murcia [9]. The *Red Book of Protected Wild Flora of the Region of Murcia* indicates the presence of *Fraxinus angustifolia* in certain locations within the Region of Murcia (Figure 3) [2,10].

**Figure 2 life-15-00163-f002:**
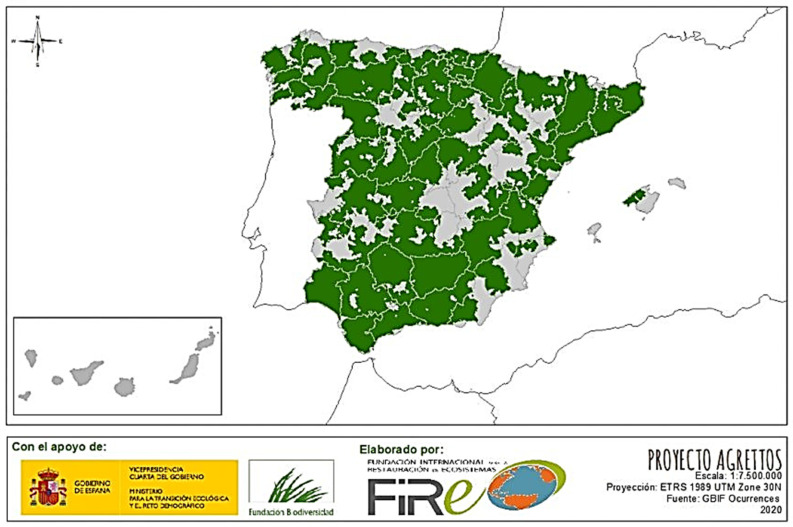
Distribution of *Fraxinus angustifolia* in Spain [9].

**Figure 3 life-15-00163-f003:**
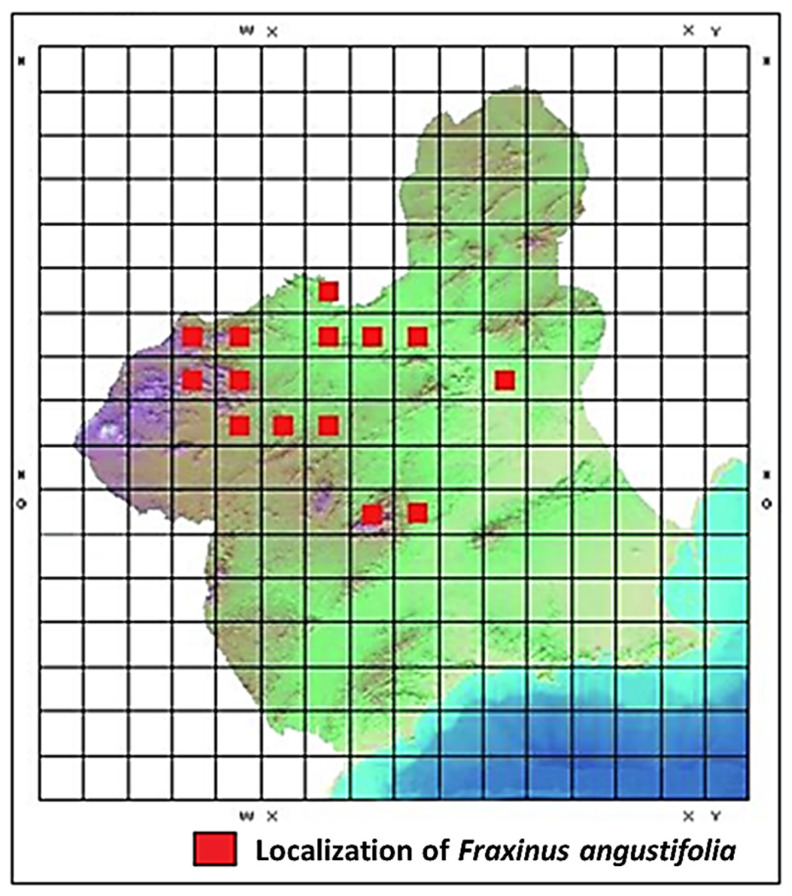
Distribution of *Fraxinus angustifolia* in the Region of Murcia [2].

The ash tree is a species from the Mediterranean and sub-Mediterranean area, which “in the Region of Murcia is scattered along the middle and upper stretches of the Segura river, in the mountains of the Northwest and in Sierra Espuña, where most of the specimens likely originate from reforestation efforts” [2]. “The narrow-leafed ash is usually associated with watercourses and areas with certain levels of moisture. Ash groves are typical formations along Mediterranean riverbanks, although they can also be found on cool slopes and in depressions with accessible groundwater”. Furthermore, “in Spain, it is present along riverbanks, groves, and forests across all provinces, although it is scarce or absent in the arid areas of the Southeast, the Cantabrian Cornice, and the mid-to-high regions of the Pyrenees” [6]. In the Mediterranean Spain, it is generally confined to river margins due to its water requirements, thriving in soils with edaphic moisture [11].

Additionally, “because it grows in the bottoms of ravines and gullies, this species is vulnerable to frequent floods, although, along the Segura valley, it is often located in the outermost bands. It is also threatened by channel restoration works. In anthropized environments, it is common to observe burned specimens as a result of agricultural cleaning activities. In less humid locations, it is affected by frequent droughts, as observed in Sierra Espuña, where it is considered a priority species. The best specimens are cataloged as singular trees (e.g., Fuentes del Marqués in Caravaca de la Cruz and Cieza)” [2].

Regarding the “habitat it occupies in the Iberian Peninsula, two types of ash groves can be distinguished: riparian or azonal ash groves, dominated by *Fraxinus angustifolia*, and hillside or zonal ash groves, dominated by *Fraxinus angustifolia* and *Fraxinus ornus*”. The former “have been largely ignored in many phytosociological studies of Iberian vegetation” [12]. It is “indifferent to soil type but, due to its water requirements, appears associated with deep alluvial soils. It tolerates waterlogging well and is resistant to both winter cold and summer heat” [13].

In some regions, “restrictions on its use and exploitation may exist. For example, the rarity of ash populations in the semi-arid Southeastern zone has led to being classified as a species ‘in danger of extinction’ in the Region of Murcia” [6]. In this region, *Fraxinus angustifolia* populations are small, confined to territories that promote the isolation of small clusters or scattered specimens throughout the region. As such, it is listed as a “species ‘in danger of extinction’” in the Regional Catalog of Protected Wild Flora of the Region of Murcia [14]. Projections for its future evolution due to climate change predict a reduction in its range in southern Spain, with estimates suggesting that it may disappear from the Region of Murcia by 2071 [11].

Furthermore, arid zones have expanded, driven by global warming. This situation has caused changes in the abundance and seasonal activity of some plant species. As a result of climate change, extreme climatic and meteorological events, such as heat waves, have become more frequent, intense, and prolonged. Similarly, the frequency and intensity of droughts are expected to increase in the future, as is already the case in some areas prone to drought, particularly in the Mediterranean. This leads to increased aridity [15].

Land areas threatened by desertification risk account for 40% of the Earth’s surface. In Spain, the Canary Islands and the Mediterranean region are the areas at greatest risk of desertification [16]. The Region of Murcia, in the southeast of the Iberian Peninsula, is one of the driest areas in peninsular Spain, with much of its territory at medium-to-very-high risk of desertification [17]. Spain has the highest drought risk in Europe, both under current climatic conditions and future projections. The Region of Murcia is among the areas most affected by climate change [18].

Desertification will impact natural systems under climate change. Increasing aridity raises the risk of extinction for some plant species, particularly those already threatened by small populations or restricted habitats [15]. Deforestation and forest degradation remain major global threats. Forests are among the largest carbon and biodiversity reservoirs on Earth and are crucial for mitigating climate change [19]. Understanding the health of terrestrial ecosystems is a complex task [20]. Studies on the impact of climate change in Mediterranean climates in Italy have shown that *Fraxinus angustifolia* and other species are affected by drought during a critical stage of their physiological development [21].

As previously stated, several authors have argued about a general or riverbed localization of *Fraxinus agustifolia* in the southeastern Iberian Peninsula and the influence of climatic factors in its distribution. This is controversial and the main focus of this work. Indeed, our working hypothesis is that thermopluviometric and edaphological characteristics are of the utmost importance in the localization and distribution of *Fraxinus angustifolia* in semi-arid regions, such as the Region of Murcia.

The classification of *Fraxinus angustifolia* as an endangered species in the Region of Murcia highlights its significance, especially given future projections of its disappearance. Studying the species across various aspects—chorological, botanical, etc.—is of interest. Investigating its botanical and health status, distribution, densities, and number of specimens could serve as potential lines of research for this species in arid and semi-arid areas. Obtaining information about the number of specimens and their distribution could be the first step in understanding the species’ situation and is the main objective of this work.

## 2. Materials and Methods

### Study Area and Methodology

The study area was the Region of Murcia, situated in the southeast of Spain (Figure 4). A survey was conducted across the entire Region of Murcia to locate various trees. The process began with references to previous records of the species’ presence. Initially, the trees described by Sánchez-Sánchez et al. (2012) [10] and those cited by Sánchez-Gómez et al. (2002) [2] were located. Subsequently, the search was extended to the rest of the Region of Murcia based on information provided by members of the Environmental Agents Corps of the Region of Murcia (Spain) and the Faculty of Biology at the University of Murcia (Spain).

Environmental conditions required by this species were also considered, so another regional criterion was to survey the banks of the Segura River and the river channels located within protected natural areas. Later, locations outside the river channels with potentially suitable climatic conditions for the species, such as water springs and mountain orchards, were also visited. The trees found were geolocated using GPS for cartographic representation. In the case of river channels, surveys were conducted along both channel banks in the form of linear transects. In other geomorphologies, the entire territory was covered using random transects, locating trees individually without establishing predefined plots in any case.

The study was framed within what is known as chorological characterization. “Chorological studies were conducted at various scales of analysis, depending on the objectives, such as understanding the distribution of a specific group of taxa in a more or less extensive area—a medium-scale analysis. In this work, the goal was to characterize a floristic group based on its distribution”. An ecogeographical approach, as the relationship between the environmental characteristics of the territory and the ecological conditions necessary for the species’ survival, was adopted to highlight the relationship between the most significant ecological characteristics of the area (climate, orography, river channels, and altitude) and the studied species, “considering any particular features of the site that are of interest for the ecogeographical characterization” of the species in question [23].

Among all the trees found, those located in parks or gardens were excluded from the study. The remaining trees were categorized as either originating from reforestation efforts but naturalized in forest contexts or as spontaneous individuals. Among the spontaneous trees, this study focused on identifying the relationships between ecological characteristics and those trees forming small clusters resembling groves.

## 3. Results and Discussion

### 3.1. Environmental Characteristics

The climatic conditions, as part of the environmental characteristics of the study area, the Region of Murcia, are diverse [24]. An average representation of these conditions over more than two decades shows an increase in temperatures and significant variability in rainfall (Figure 5).

The results obtained in the field show the existence of scattered trees across the regional territory and small wooded patches. These are found in Sierra Espuña (Casa Rosa and Casa Leyva—the latter being introduced species) and in the municipality of Bullas (Estrecho de Capitán, Arroyo del Chaparral, and Rambla de la Regidora or Asomadilla). Their comparative climatological data are shown in Table 1

In the area of the municipality of Bullas, “the average annual temperature ranges between 15 and 16 °C, with no monthly average falling below 7.5 °C. Frost days in the municipalities of Bullas and Cehegín range between 5 and 10 per year. Days when the temperature exceeds 30 °C are approximately 50. Frost events occur 95% of the year and can extend into the second ten days of April, with a probability of 5%. The average annual precipitation varies between 300 and 375 mm, while potential evapotranspiration ranges between 800 and 830 mm. Precipitation peaks occur in October and April and the soil water deficit period lasts 4 to 5 months. The aridity index ranges from 53 to 55” [25,26,27].

The climatic conditions of Sierra Espuña show an “average annual temperature of approximately 13.5 °C. Temperatures drop below 0 °C on about 40 days per year and exceed 30 °C around 50 days. These values correspond to an altitude of about 900 m, with colder conditions at higher altitudes and warmer temperatures at lower altitudes. Annual precipitation is approximately 470 mm, with a potential evapotranspiration of 745 mm at an altitude of 800 m. The precipitation maximum occurs in October, with a secondary, albeit smaller, peak in April. The dry period is limited to the summer months, and there may even be an excess of soil water in April. The aridity index is close to 50” [28].

Other authors report that in the “Northwestern quadrant” of the region, in the “sector with altitudes below 750 m” where the Bullas masses are located, “the average annual temperature ranges between 14° and 16 °C, the frost-free period lasts around 7 months, and the average annual precipitation is 400 to 450 mm/m^2^” [28]. Additionally, “toward the more eastern part of the region (Bullas and Cehegín), the soil water deficit period increases to 4–5 months, extreme temperatures are less pronounced, and the frost-free period is longer”. Similarly, in Sierra Espuña, “at higher altitudes (≥1000 m), there are more than 40 frost days per year, precipitation exceeds 480 mm, and the dry period is limited to the summer months…” [29].

### 3.2. Territorial Distribution

As early as 1751, the presence of ash trees in the Region of Murcia was documented [4], although their exact geographic location was not specified. Currently, they can be found in various locations, with the majority located along ephemeral riverbeds and with continuous flows over time. The ash trees are absent in coastal areas as well as nearby zones, the northwesternmost area, and the northern part of the region, with some exceptions in the latter case (Figure 6).

The presence of trees in relation to the distribution of the region’s average annual rainfall shows their absence in areas with lower precipitation. The trees located near the city of Murcia are found along the banks of the Segura River, where water availability is high despite being in a low-rainfall environment. Additionally, it should be noted that these trees are the result of artificial reforestation efforts but have achieved a high degree of naturalization. Notably, the species is absent in the rainier areas of the region, such as the north and northwest, due to other environmental factors (Figure 7).

### 3.3. Interrelation Between Ecological Factors and the Species

Our fieldwork resulted in the location of 670 trees, as isolated individuals or as part of small wooded groups. Some of the trees resulting from reforestation in natural and/or semi-natural environments, which have become naturalized, show adequate development as they are located in areas with water availability, such as rivers, ravines, and similar locations. A total of 195 trees originating from reforestation and 475 trees from spontaneous-natural generation were recorded, of which 221 are associated with a spring and 254 with river channels. The latter are distributed across various groves throughout the regional territory. An exception is the Luchena River (Lorca, in the western part of the region), where the trees introduced into the riverbed failed to survive.

All terrain features modify ecological conditions, transcending mere physiognomy to form the foundation of ecological systems [32]. Clear examples are riverbeds, which ash trees utilize for their existence. Ash trees, like other living organisms, have specific ecological requirements, with water availability being the most critical factor. The optimal rainfall conditions for their proper development are annual precipitation levels equal to or exceeding 750 mm/m^2^ (Table 2). This optimal requirement is not met in the study area, although some locations in the northwest approach this rainfall regime (Figure 7). This shortfall is compensated by the trees’ location in riverbeds, where the water table is closer to the surface, providing more favorable ecological conditions facilitated by the terrain [32]. For this reason, the majority of the trees were found in riverbeds, except for a group located near a spring in Sierra Espuña with 221 trees.

Regarding temperatures, the entire region is suitable for their existence except in areas where the average annual temperature is below 7.5 °C, which does not occur in the study area. Only the peaks of Mount Revolcadores, at an altitude of nearly 2100 m, approach the temperature limit with an average annual temperature of 8 °C.

In terms of altitude, the trees found are located below 1100 m, with the group in Sierra Espuña near the spring being situated at this elevation.

Studies conducted on ash groves (*Fraxinus angustifolia*) in the mountains of the Community of Madrid, in the central Iberian Peninsula, have demonstrated the phreatophytic nature of the species, as 74.71% of the soil surface where the trees were located is classified as having medium-to-high moisture. The morphostructural units on which the trees are found are 60% in piedmont areas and the rest in depressional zones. A total of 76.50% of the ash groves prefer flat terrain or areas with a slight slope (<10%). Regarding bioclimatic zones, the distribution of the ash groves is as follows: 29.09% in the Mesomediterranean zone, located between 700 and 800 m in altitude, and 70.38% in the Supramediterranean zone, at altitudes between 800 and 1500 m [35].

The ash trees in the Region of Murcia also exhibit this phreatophytic character, being restricted to river channels where groundwater moisture is higher, due to the surrounding arid environment. They are unable to occupy other structural geomorphologies with less slope. The bioclimatic zone where the trees in the Region of Murcia are located is the Mesomediterranean, at altitudes ranging from 130 to 1120 m.

In the Moroccan Atlas, holm oak forests give way to *Fraxinus dimorpha* in more arid areas, contrary to what has been observed in the study area of Murcia, where ash groves coexist with river channels while external arid zones are exclusively occupied by holm oak forests, never by ash groves [36].

In well-developed trees, the presence of certain herbaceous and/or shrubby species associated with the ecological conditions created by the ash trees has been detected, similar to what occurs in the ash trees studied in Madrid [35].

Since water availability is one of the most limiting factors for the proper development of plant species—particularly in highly arid zones such as the study area—and given the water requirements of the species, it may serve as an indicator of climate change in response to aridity (Table 2).

## 4. Conclusions

The increasing aridity, combined with the environmental requirements, makes *Fraxinus angustifolia* a very sensitive species in the south of Spain. The interest of this study lies in the geolocation of the 670 trees, the identification of the natural or semi-natural environments where they are found, and their regional distribution. It is noteworthy that the species is entirely absent in the high-altitude northwestern areas of the Region of Murcia, despite the favorable conditions for its existence, especially considering that the species survives under similar conditions in the northern mountains of Madrid. Regarding the proposed hypothesis, it must be stated that the species is absent from the entire southern area of the Region of Murcia, as well as from much of the northwestern and northern areas. In addition, out of the 475 trees from spontaneous-natural generation, 221 are located near a spring, while the remaining 254 trees are found in river channels. However, these trees form small and scattered groups, which poses a significant challenge to their conservation against both natural and human-induced adversities.

The sites where the species is found, whether originating from reforestation or spontaneously, maintain a groundwater level that supports its presence, making them potential reservoirs for its preservation in the face of climate change. At the same time, *Fraxinus angustifolia* can serve as an indicator of soil water availability in relation to groundwater levels, aquifer usage, and the effects of climate change on water reserves. Therefore, monitoring its reduction, expansion, and development could provide insights into how climate change is evolving and the potential extinction of this species in arid and semi-arid regions.

## Figures and Tables

**Figure 1 life-15-00163-f001:**
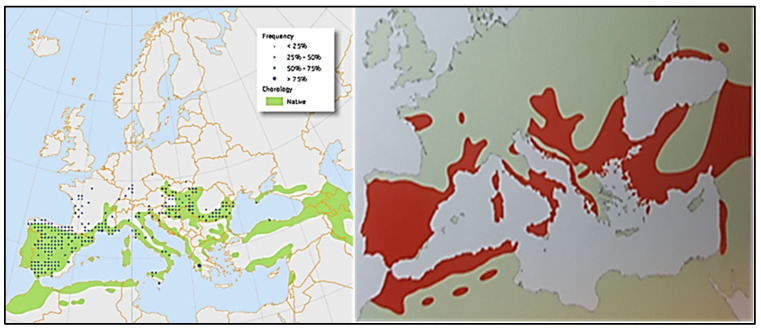
Distribution of *Fraxinus angustifolia* in Europe and the north of Africa. Adapted from Musel et al. (1978) cited in [1,2].

**Figure 4 life-15-00163-f004:**
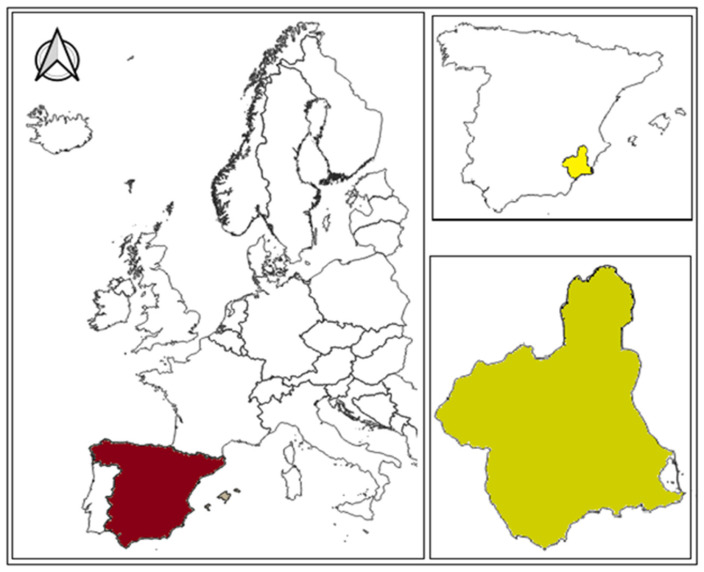
Location of the study area [22].

**Figure 5 life-15-00163-f005:**
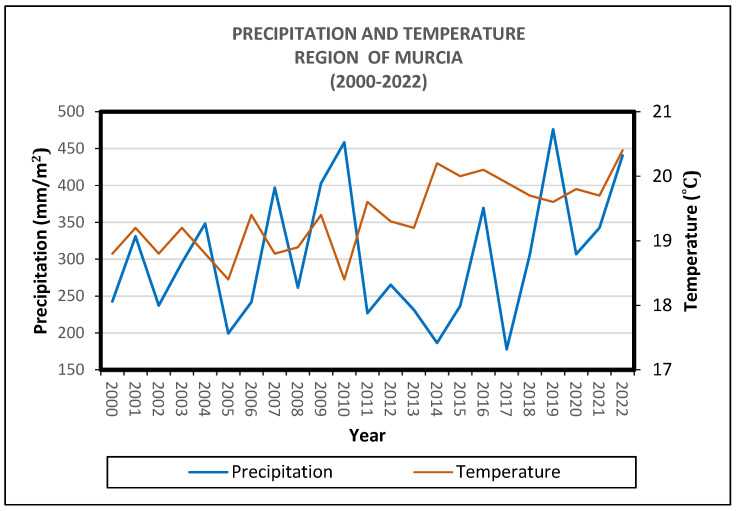
Evolution of temperatures and rainfall, Guadalupe weather station (Region of Murcia) [1].

**Figure 6 life-15-00163-f006:**
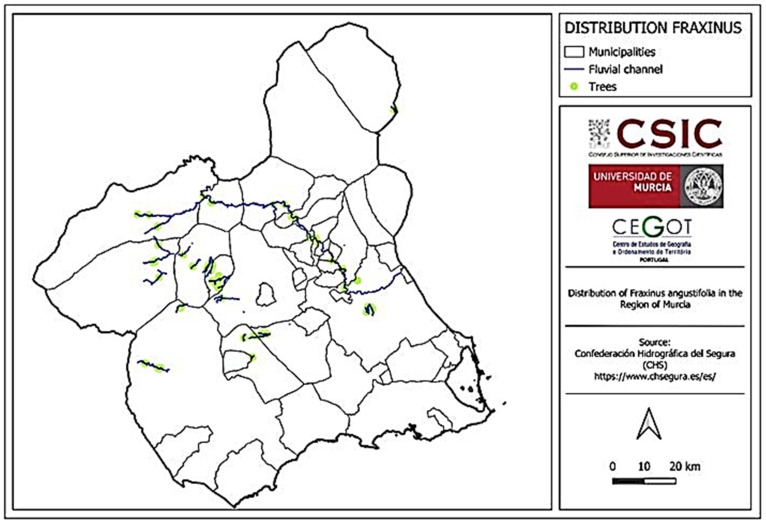
Distribution of ash trees in the Murcia Region associated with fluvial channels. Information obtained from https://www.chsegura.es/es/ (access on 18 November 2024) [30].

**Figure 7 life-15-00163-f007:**
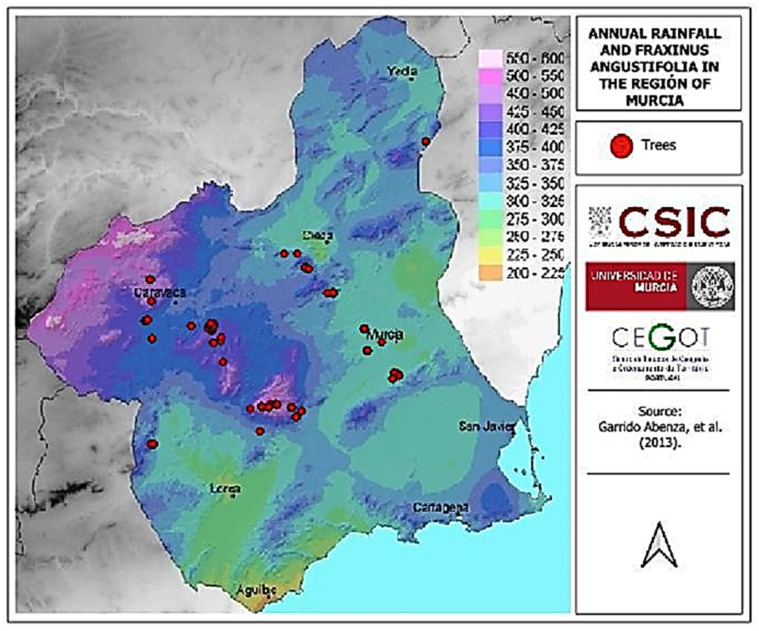
Distribution of annual rainfall and *Fraxinus angustifolia* in the Region of Murcia obtained from Garrido-Abenza et al (2013) [31].

**Table 1 life-15-00163-t001:** Climatological data according to various authors in the area of Bullas and Sierra Espuña.

Area	Parameters	Albadalejo Montoro [4]	González Ortiz [5]	Conesa-García and Alonso-Sarria [6]
BULLAS	Average annual temperature (°C)	15–16	15.4	14–16
	Rainfall (mm/year)	300–370	377	400–450
	Days of frost	5–10	---	---
	Days ≥ 30 °C	50	---	---
SIERRA ESPUÑA	Average annual temperature (°C)	13.5	---	---
	Rainfall (mm/year)	470	---	480
	Days of frost	40	---	40
	Days ≥ 30 °C	50	---	---

**Table 2 life-15-00163-t002:** Ecological requirements for *Fraxinus angustifolia* [33,34].

	Optimal conditions					Limiting conditions						**Comments**
	Very limiting conditions					Non-adequate conditions						Ahs does not tolerate severe drought. It can withstand a rainfall deficit, provided it can capture groundwater to compensate, such as near rivers or in areas with an accessible water table.
Annual rainfall (mm)												
400-	450	500	550	600	650	700	750	800	850	900	950+	

Annual mean temperature (°C)												Cold (and therefore altitude) limits the growth of trees by shortening the number of months in which they can grow. Ash trees can withstand extreme cold in Winter (when they are leafless), but they are sensitive to spring frosts, which can damage the terminal guide and cause the formation of forks, which must be corrected during training pruning.
6-	6.5	7	7.5	8	8.5	9	9.5	10	10.5	11	11.5+	

Altitude (m)												
150-	300	450	600	750	900	1050	1200	1350	1500	1650	1800+	


## Data Availability

All data will be available under reasonable request to the corresponding author.

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
