# Peer review of "Environmental Factors Determining the Chorology of Fraxinus angustifolia in the Region of Murcia (Southeastern Spain)"

_life, 2025, doi:10.3390/life15020163_

Round 1
Reviewer 1 Report
Comments and Suggestions for Authors
Please refer to my concerns/comments/queries written in the text boxes embedded in the manuscript itself.

Author Response
Please, see the responses to the comments included in the attached PDF of the article.

Reviewer 2 Report
Comments and Suggestions for Authors
Evaluation:
1. The objectives of this paper are clearer, the methodology and content statement are more complete, and this correlation analysis is of high relevance.
2, the professionalism and scientificity of the article is also good, the core content and research methods and results are more clear
Recommendation:
1. Although it may be more interesting for scholars who conduct research on tree species, it is a bit monotonous in terms of the whole environmental research, and it does not find the ecological role of the tree as well as its importance in the article, so the persuasive power seems to be not enough.1.
2. The article says it provides insight etc. but I didn't find enough of an in-depth exploration of the rest.
3. There seems to be room for more in-depth analysis of the distribution of tree characteristics
Author Response
Please, see the attachment.

Round 2
Reviewer 1 Report
Comments and Suggestions for Authors
Now the manuscript is significantly improved.